# Insights into the Impact of Physicochemical and Microbiological Parameters on the Safety Performance of Deep Geological Repositories

**DOI:** 10.3390/microorganisms12051025

**Published:** 2024-05-19

**Authors:** Mar Morales-Hidalgo, Cristina Povedano-Priego, Marcos F. Martinez-Moreno, Miguel A. Ruiz-Fresneda, Margarita Lopez-Fernandez, Fadwa Jroundi, Mohamed L. Merroun

**Affiliations:** Department of Microbiology, Faculty of Sciences, University of Granada, 18071 Granada, Spain; ppriego@ugr.es (C.P.-P.); mmartinezm@ugr.es (M.F.M.-M.); mafres@ugr.es (M.A.R.-F.); margaritalopez@ugr.es (M.L.-F.); merroun@ugr.es (M.L.M.)

**Keywords:** nuclear waste, radiation, bentonite, corrosion, microorganism, compaction, temperature, deep geological repository

## Abstract

Currently, the production of radioactive waste from nuclear industries is increasing, leading to the development of reliable containment strategies. The deep geological repository (DGR) concept has emerged as a suitable storage solution, involving the underground emplacement of nuclear waste within stable geological formations. Bentonite clay, known for its exceptional properties, serves as a critical artificial barrier in the DGR system. Recent studies have suggested the stability of bentonite within DGR relevant conditions, indicating its potential to enhance the long-term safety performance of the repository. On the other hand, due to its high resistance to corrosion, copper is one of the most studied reference materials for canisters. This review provides a comprehensive perspective on the influence of nuclear waste conditions on the characteristics and properties of DGR engineered barriers. This paper outlines how evolving physico-chemical parameters (e.g., temperature, radiation) in a nuclear repository may impact these barriers over the lifespan of a repository and emphasizes the significance of understanding the impact of microbial processes, especially in the event of radionuclide leakage (e.g., U, Se) or canister corrosion. Therefore, this review aims to address the long-term safety of future DGRs, which is critical given the complexity of such future systems.

## 1. Introduction

The increasing production of radioactive waste due to the extensive use of nuclear power has underscored the urgency for developing reliable strategies for the long-term containment of these hazardous materials. In response to this emerging challenge, the deep geological repository (DGR) model has been proposed for the secure confinement of radioactive waste [1]. This system entails depositing nuclear waste in metallic canisters underground at a depth of approximately 500 m within stable geological formations. Depending on the country, the materials used in the various barriers, as well as other physico-chemical characteristics related to the design of the repository, may vary. Table 1 presents the main information concerning the materials and their properties to be expected in future repositories of the principal national companies involved in the development of DGRs. In many countries including Spain, Switzerland, Belgium, Finland, France, and Canada, the canisters will be surrounded by compacted clay materials selected for their ability to provide mechanical, hydraulic, and thermal protection [2,3]. Specifically, bentonite clay has been recognized as a suitable material to use as both artificial and natural barriers within the DGR systems due to its exceptional properties. The utilization of bentonite from various locations has been extensively investigated as a buffer material. Notable examples include MX80 from the United States, FEBEX from Spain, FoCa from France, and GMZ from China. These formations particularly exhibit low permeability (resulting in a decrease in groundwater filtrations), mechanical support (ensuring stability), swelling capacity (facilitating the self-sealing of cracks), thermal conductivity (preventing overheating), and ion exchange capacity (enabling radionuclide retention) [4]. In addition, bentonite presents optimal compaction properties, which not only significantly contribute to improving the mechanical support, thermal conductivity, and sealing properties mentioned previously, but may also influence the viability of the bentonite microbial communities. Highly compacted bentonite blocks are thought to prevent microbial growth [5]. Despite this, recent studies have demonstrated microbial stability in compacted bentonite under conditions that mimic a DGR [6]. Thus, these microorganisms are expected to maintain their viability and metabolic activity within such a harsh system.

Microorganisms have the potential to affect their surrounding environment and, consequently, the safety of DGRs through various processes. These include the generation of gases, corrosion of metal canisters [20], alteration of redox conditions [21], transformation of mineral clays [22], and interaction with radionuclides [23,24]. A wide variety of physical (temperature, radiation, groundwater filtration, bentonite compaction, etc.), chemical (gaseous compounds, corrosion, presence/absence of oxygen, etc.), and biological factors (microbial activity) may compromise the performance of both bentonite barriers and their microbial communities. Given all these considerations, evaluating how these factors may compromise the safety of these storage facilities is of crucial importance to ensure the secure disposal of these highly polluting and hazardous waste materials, which threaten the health of living organisms.

This review provides a comprehensive analysis of the most recent advances in studies predicting and evaluating environmental conditions that would affect the performance and safety of future deep geological repository systems. A special focus is given to the effect of radiation, temperature, bentonite compaction and various biotic and abiotic factors on the microbial behavior in bentonite barriers.

## 2. Effect of Radiation

The DGR concept is designed almost exclusively for the management of so-called high-level waste (HLW). HLW includes radioactive waste with substantial amounts of long-lived alpha and beta–gamma emitters, making this waste highly radioactive and capable of generating high temperatures. Most of this type of waste includes spent nuclear fuel (SF) generated by nuclear power plants. The other minority contains waste resulting from the reprocessing of SF, as well as residues from research, industry, and mining [25,26]. One of the main concerns regarding HLW is the presence of elements with very long half-lives, spanning hundreds of thousands of years [27]. Hence, examining the impact of radiation, particularly gamma radiation due to its high penetrative ability, on different barriers of future repositories has been of paramount importance from the outset to guarantee long-term safety. The measurement of radiation energy absorbed by a material is quantified as the absorbed dose, which is expressed in the international system of units (SI) as Gray (Gy). Ionizing radiation can interact with matter through various mechanisms and thus alter its properties [28]. This section will only focus on the effects of gamma radiation on copper (Cu) corrosion, bentonite stability, and microbial presence and survival. In general, the absorbed doses at the surface level of the canister are expected to be relatively low, though this will vary depending on the DGR model (Table 1). Furthermore, as the distance from the radiation source increases, the received radiation doses decrease progressively. Consequently, the radiation levels experienced within the canister will be higher than those within the engineered bentonite barrier, and the radiation within the latter will be higher than that received by the host rock. Despite the relatively low radiation doses experienced in the various barriers, some studies have been conducted on the impact of gamma radiation on canister corrosion and the bentonite barrier to prevent potential leaks.

### 2.1. Copper Corrosion

Gamma radiation primarily arises from the decay of the fission product Cs-137, which possesses a relatively short half-life of 30 years. Consequently, the intensity of this radiation is expected to decrease relatively quickly following the closure of the repository [29]. Its impact on copper corrosion would be mainly indirect, occurring through its reaction with water and the subsequent decomposition into oxidizing (HO· and H_2_O_2_) and reducing agents (H·, and H_2_), which is known as the radiolysis process [30]. Of all the species generated, the most concerning are hydrogen peroxide (H_2_O_2_) and the hydroxyl radical (HO·), as they are the most thermodynamically stable and have higher standard reduction potential than copper [31]. In addition, radiolysis of humid air can also generate other reactive species such as NO_x_ and HNO_3_ [32]. Björkbacka et al. [33] reported that absorbed doses of gamma radiation caused corrosion in copper under conditions of anoxic aqueous solutions, both uniformly and locally. The main corrosion products were copper (I) oxide (Cu_2_O) along with a small fraction of copper (II) oxide (CuO). In recent years, several studies have been focused on examining the corrosion products on the surface of copper canisters following exposure to gamma radiation doses. However, a challenge persists as most of these studies have been conducted at doses significantly higher than those expected in the repository. This discrepancy arises from the challenge of conducting experiments under realistic repository conditions, where radiation exposure rates will be very low but extended over time. Consequently, the corrosion magnitude observed in these studies surpasses that predicted by simulation models [34,35,36]. The research by King et al. [37] demonstrated that to induce significant corrosion concerns, a threshold of 100 Gy/h would need to be surpassed. Additionally, they suggested that dose values below this threshold could marginally decrease corrosion, possibly due to the formation of a passive protective oxide layer that many metals develop to resist corrosion.

### 2.2. Bentonite Stability

Due to the dispersal capability of gamma radiation, potential effects could occur on the buffer barrier as well [38,39]. This review will specifically focus on its impact on the chemical and mineralogical composition of bentonite clay, as well as its cation exchange capacity [39]. Therefore, one of the essential requirements for its utilization as a buffer and sealing material would be the rheological and chemical stability when exposed to ionizing radiation and even in the presence of radionuclides in the worst-case scenario of a waste leak [40,41,42]. Radiation can impact the clay structure and, consequently, its properties, through processes such as inducing amorphization and affecting the sorption capacity of elements. This would compromise the long-term stability of this material and, thus, the safety and integrity of the repository [43]. However, most experiments clearly indicate that the amorphization of bentonite can be ruled out under radiation levels anticipated in future DGRs, as this phenomenon only occurs at significantly higher levels. For instance, Sorieul et al. [44] reported that a radiation dose of at least 1011 Gy is required for clay to become amorphous. Additionally, Galamboš et al. [38] concluded that bentonite from Slovakia exhibited no changes or very insignificant ones in their structural properties after exposure to gamma radiation. Nevertheless, in a study conducted in 2011 by Holmboe et al. [45], the effects of γ radiation on the ability of MX80 bentonite to retain radionuclides were tested. They observed a notable decrease in Co_2_^+^ sorption in irradiated samples but not in Cs^+^ which may suggest that this form of irradiation could have altered the surface characteristics relevant to the sorption of this radionuclide. Additionally, bentonite is notable for its high content of accessory minerals. It has been observed that the distribution of these minerals could also impact the behavior of this clay, as they may dissolve and alter the properties under radiation conditions. On the other hand, other studies have demonstrated that the stability of bentonite against radiation depends on the water content, attributed to the radiolysis effect occurring within its pores. Gu et al. [46] noted that the stability of bentonite increases when the water content is lower, as it results in a lower number of products from the radiolysis of water. More recent studies, such as that of Chikkamath et al. [47], have corroborated this effect by finding that doses of up to 12 kGy/h had no effect on Fe(II)-clay powder samples. This outcome was probably due to the fact that the irradiated samples were dry, resulting in a lower impact of water radiolysis. Indeed, it has already been highlighted the contradictory results concerning the alterations in the surface reactivity of bentonite when subjected to radiation [39]. Whilst some authors have reported an increase in the ion exchange capacity, others have observed the opposite effect. Accordingly, current knowledge continues to advance in this field to establish the impact of this radiation on buffer clay materials.

### 2.3. Microbial Viability

Radiation would also affect the microbial communities present in the future repositories by altering their structure, viability, and activity. To compare radiation sensitivity of microorganisms, the term decimal reduction dose (D10) is used, which is defined as the radiation dose (in kGy) necessary to reduce a microbial population by 90% of its total number [48]. As previously mentioned, the effect of radiation depends on the perceived dose, which will vary depending on various factors including the distance from the radiation source, the barrier model, the type of radionuclide contained, and its lifetime [49]. Not only does radiation itself cause changes and damage at the cellular level, but the reactive oxygen species (ROS) resulting from the radiolysis process will also affect most biomolecules. One of the most prominent forms of damage is to DNA, causing DNA strand breaks, base changes, mutations, etc. [50]. In general, microorganisms respond to such damage with different defense mechanisms to repair as much as possible. These mechanisms include DNA repair systems, production of antioxidant enzymes to cope with reactive oxygen species (ROS), and antioxidant processes involving increased intracellular concentrations of inorganic solutes and pigments [51]. The response of microorganisms to radiation varies amongst strains, as sensitivity depends on factors such as cellular water content, DNA size and structure, antioxidant, and DNA repair systems, as well as the ability to develop resistant cell forms, amongst others. The latter is particularly important, as several studies have shown that the tolerance of bacterial spores to radiation is significantly higher than that of vegetative cells [48]. Furthermore, this sensitivity also depends on environmental conditions: water activity, level of desiccation, presence or absence of oxygen, etc. For instance, many studies reported that conditions such as a dry environment, lack of oxygen, and low temperatures enhance radiation resistance [52,53,54]. The literature on bacterial species highly resistant to radiation is quite limited; however, two strains are referenced and have been well studied for their high resistance to radiation, namely *Deinococcus radiodurans*, with a tolerance of up to 17 kGy [55], and *Kineococcus radiotolerans* [56,57]. In general, the majority of strains capable of resisting certain doses tend to be extremophiles, exhibiting tolerance to other extreme conditions of temperature, desiccation, or high concentration of salts [49]. On the other hand, regarding the bentonite bacterial community, some studies have reported *Bacillus*, *Acinetobacter*, *Desulfosporosinus*, and *Clostridium* to also be resistant to radiation doses [58,59].

Nevertheless, in the future repositories, microorganisms will not only face radiation as a stress factor. Therefore, more research is needed to understand the evolution of microbial communities under the combination of various repository conditions.

## 3. Effect of Bentonite Dry Density and Microbial Activity on Bentonite Performance as an Engineered Barrier in DGRs

HLW will be contained within canisters crafted from corrosion-resistant materials such as copper, stainless steel, or “novel materials”, which are typically subjected to very low corrosion rates. Regarding novel metal canisters, the advanced coatings and alloys (e.g., Ni or Ti alloy and ceramic coated metals) are expected to delay corrosion, but their degradation is not ruled out when subjected to hydration, irradiation, and temperature. The different metallic canisters will be encased in a protective buffer of compacted bentonite securely stored deep underground, at depths of several hundred meters, thus ensuring a robust containment system for the safe disposal of the nuclear waste [60].

Bentonite is rich in a swelling mineral, typically montmorillonite, which is a key component with a notable swelling ability [61,62]. Montmorillonite, belonging to the smectite mineral group, acts as an ion exchanger. It features a stable negatively charged silicate layer with an interlayer hosting mobile counter cations and water molecules. The predominant counter ions are often Na^+^ and Ca^2+^, resulting in Na- and Ca-bentonite, respectively, although other ions may also be present [63]. High quality commercial bentonite typically consists of over 80% montmorillonite. However, this content varies significantly amongst different commercial bentonites, ranging from 60% to more than 80% [64]. During the canister deposition process, the bentonite buffer is composed of low-water-content (10–17%) bentonite blocks. Strategic slots will be positioned between the bentonite, the canister, and the rock, enabling a smooth lowering of canisters and blocks into deposition holes. Upon contact with the groundwater, bentonite will expand and undergo mechanical pressure until it attains the intended full compaction density of 2 kg/m^3^ and a relatively low water content of approximately 26%, leading to the sealing of the repository [6]. In this way, the buffer materials will allow an effective sealing of any potential radiation leakage pathways and maintain the mechanical integrity of the canister, thereby enhancing the safety of the DGR [61,65]. Consequently, bentonite has emerged as the preferred choice for the final setup of the disposal due to its advantageous mechanical support, ensuring stability for canisters. Additionally, its low permeability mitigates groundwater infiltration, whilst its high ion exchange capacity aids in the retention and retardation of radionuclides in case of a system breach [61,66,67]. Furthermore, its high plasticity and swelling capacity facilitate the self-sealing of canister cracks, whilst its good thermal conductivity and optimal properties for compaction enhance its suitability for repository applications [4,68,69]. Across different waste disposal concepts, the target swelling pressure is set at a minimum of 5 MPa, requiring a dry clay density of >1.6 g/cm^3^ [70]. A primary concern associated with the bentonite buffer involves the potential interaction of microorganisms with minerals, leading to weathering, dissolution, and the formation of secondary minerals [71]. A high bentonite density is thought to exert an inhibiting effect on the activity of the natural bacterial populations within bentonite clay [6,72]. The inhibition of bacterial growth and the spore germination is likely due to limited pore space, low water activity, and high swelling pressure. These findings that are supported by many previous studies [71], indicating the restricted microbial activities under 1.6 g/cm^3^ or higher, were further corroborated by the low extractability of solid-phase bentonite natural organic matter [72]. Although these studies were conducted for a duration of up to 18 months, the authors extrapolate their results to the expected time frame required for a DGR to reach full saturation, which ranges from 50 to 5000 years [71]. These outputs suggest that bentonite compacted to densities higher than this threshold may exhibit a long-term greater stability and has significant practical implications for the safety assessment of the DGRs.

Nevertheless, there are many prokaryotes adapted to survive in high pressure environments, as well as the endospore of any spore-forming bacteria, which have been reported to not completely disappear after 15 months under the repository conditions [73]. Therefore, a continuous evaluation of microbial activity and diversity in such a challenging environment is essential for the safety assessments of the DGR concept. In recent years, numerous studies have investigated the behavior of allochthonous bacteria in highly compacted bentonite (Figure 1), as well as the cultivation of microbes derived from the bentonite itself [6,64,74,75]. For instance, Jalique et al. [76] identified Gram-positive spore-forming bacteria in highly compacted bentonite and inferred that the formation of spores might enhance bacterial survival in the challenging conditions of this environment.

In addition, according to Bengtsson and Pedersen [77], a high density of bentonite buffers in future DGRs will significantly reduce the risk for sulfide production in the buffer and the concomitant corrosion of copper canisters by limiting microbial activity. However, even under these harsh conditions, certain anaerobic bacterial groups, notably sulfate-reducing bacteria (SRB) and iron-reducing bacteria (IRB), have been reported with the potential to be active within the harsh conditions of highly compacted bentonite [6,73]. The adverse effects of their activity encompass the initiation of microbiologically influenced corrosion (MIC) processes, the conversion of Fe(III) to Fe(II) in smectite (the primary mineral in bentonite), and the dissimilatory reduction in sulfate, thiosulfate, and sulfur to sulfide by sulfide-producing bacteria (SPB) as a main concern for the safety of a geological disposal, since sulfide is a corrosive agent for metal waste canisters, and, in particular, for copper canisters [78]. In line with this, Povedano-Priego et al. [75] reported that high bacterial diversity was detected in the acetate-treated Spanish bentonite compacted at 1.5 and 1.7 g/cm^3^ densities after 24 months of anoxic incubation. Amongst the identified microorganisms in the highly compacted bentonite, there were bacteria involved in the sulfur (e.g., *Desulfuromonas* and *Desulfosporosinus*) and iron (e.g., *Thiobacillus* and *Rhodobacter*, *Geobacillus*) biogeochemical cycles, as well as those (e.g., *Delftia* and *Stenotrophomonas*) enriched by the presence of acetate, as the electron donor. In addition, in their study, Martinez-Moreno et al. [68] reported on the survival of SRB in compacted Spanish bentonite (1.7 g/cm^3^) with a copper disc placed in the core, after one-year anoxic incubation at 30 °C, although their growth was stimulated by the presence of electron donors (lactate and acetate) and sulfate (electron acceptor). However, in a more realistic scenario, simulating a post-closure DGR phase, highly compacted bentonite blocks (1.7 g cm^−3^) with high-purity copper disks in the core, and incubation at a high temperature (60 °C), the number of SRB was drastically reduced, even when electron donors/acceptor were added to the system, and only *Pseudomonas* as well as some bacterial groups adapted to extreme conditions were able to survive [69]. A step forward to the improved understanding of the influence of higher temperatures on the bentonite bacterial communities and radionuclide migration through the engineered barriers is crucial to determine the validity of bentonite buffer safety functions.

Of particular concern is the biotransformation of smectite into illite through Fe(III) bio-reduction, which stands out as one of the most alarming processes [79]. An attack of iron-reducing bacteria to the ferric iron component within bentonite buffers is expected to diminish the swelling capacity of the clay [65]. This, in turn, may create conditions that stimulate the microbial activity within the buffer, enhancing the diffusion of sulfide and potentially leading to the release of radionuclides [80,81]. Nevertheless, many studies have demonstrated that no illitization process was detected in highly compacted bentonite under different conditions relevant to DGRs, and different temperature incubation (both at 30 °C and 60 °C) for one year at anoxic conditions, thus confirming the mineralogical stability of the bentonite and its crucial role as effective barrier for future DGRs [68,69,75].

Additional research involving extended incubation periods (e.g., spanning at least 10 years), providing suitable electron donors (e.g., lactate and acetate) alongside terminal electron acceptors such as sulfate and Fe(III) are imperative to resolve remaining key issues about the roles of SRB and IRB and whether they can accelerate the degradation of bentonite-based buffers and thus threaten, the safety of the whole repository system. In general, conducting both short- and long-term experiments, along with modeling efforts, is essential to showcase the resilience of the DGR waste management concept. These endeavors are crucial for enhancing comprehension and predictability regarding the influence of fundamental processes and their interconnections. Such attempts pave the way for advancing our understanding throughout the long-term management of radioactive waste.

## 4. Effect of Radionuclides on the Diversity and Viability of Bentonite Microbial Communities

HLW is primarily composed of spent fuel from nuclear power plants, which is the most hazardous residue due to its radioactivity, which may persist for up to 1000 years. This waste contains a high number of radionuclides including uranium (in the form of enriched UO_2_ pellets with ^235^U), transuranic elements (plutonium and minor actinides), fission products (e.g., Be, Ce, and Se), and activation products (e.g., Ni, Mo, and Sr) [82]. Hence, the radionuclides present in HLW can pose challenges to the environment and public health due to their emission of radioactivity, in the form of beta particles.

The mobility of uranium in the environment depends on its speciation and redox state. In natural environments, U(VI) under aerobic conditions exists as UO_2_^2+^, hydroxyl complexes, and uranyl carbonate at pH below 2.5, 6.5, and 7, respectively [83]. The oxidized forms of U carry a higher number of positive charges, leading to increased solubility and mobility, thereby increasing its toxicity towards microbial cells. Conversely, U(IV) remains insoluble and exhibits lower toxicity in anaerobic environments [23]. The toxicity of selenium also correlates with its oxidation state, with oxyanions (selenate [Se(VI)] and selenite [Se(IV)]) representing the most toxic forms of Se due to their high solubility and mobility, causing harmful effects in the environment. On the other hand, metallic selenium [Se(0)] and selenides [Se(-II)] in terrestrial and aquatic ecosystems exhibit low solubility and mobility, thus presenting lower toxicity levels [84,85].

All the scenarios contemplated for the DGRs suggest that the release of radionuclides, during the long-term repository period, may be inevitable due to natural evolutionary processes and water contacting the source term. The consequence is that the radionuclides could migrate through the repository barriers and ultimately reach the biosphere [86]. The leakage of radionuclides from the canisters to the bentonite (the engineered barrier) could impact the diversity and viability of the microorganisms naturally inhabiting this material. As such, these microorganisms may influence the speciation and mobility of the radionuclides, thus limiting their migration into the biosphere.

### 4.1. Uranium

Most of the toxicity associated with uranium is due more to its chemistry as a heavy metal rather than its radiotoxicity. The toxic effects vary among plants, animals, and microorganisms, consequently leading to differing mechanisms of action [87]. In microorganisms, uranium toxicity results in reduced cell viability and metabolic activity, along with increased DNA damage. Additionally, such toxicity has been associated with elevated membrane permeability, oxidative stress, and temporary RNA degradation [88]. However, microorganisms inhabiting metal-contaminated environments may possess an increased tolerance to heavy metals and radionuclides and develop diverse mechanisms for their immobilization such as biosorption, biomineralization, bioreduction, and bioaccumulation [89].

Povedano-Priego et al. [90] demonstrated the effect of uranium on the microbial diversity of various bentonite microcosms treated with uranyl nitrate and glycerol-2-phosphate (G2P). In those microcosms incubated under aerobic conditions, significant U effects were observed, leading to the enrichment of microorganisms with ability to immobilize U through U phosphate biomineralization process, given the aerobic condition of the environment. Phosphatases play a key role in the precipitation of uranium. Production of inorganic phosphates by intracellular phosphatase, stimulated by U(VI), enhances the immobilization of uranium and its precipitation as uranium phosphates, thereby reducing the toxicity of U(VI) to cells [90,91]. Interestingly, *Amycolatopsis* was identified in high abundance in G2P-uranium-treated bentonite microcosms. This bacterium has been proven to efficiently immobilize U as uranium phosphates through phosphatase activity induced by the presence of G2P (Figure 2; [91]). Additionally, these actinobacteria can remove uranium through its biosorption by carboxyl, amide, and hydroxyl groups [92]. Similarly, *Bacillus*, a bacterial genus renowned for its biomineralization and uranium biosorption capabilities, has been identified in the uranium-treated bentonites [93]. For instance, Merroun et al. (2005) reported that the S-layer, a protein envelope encasing *Bacillus sphaericus* JG-A12, exhibited the capacity to sequester uranium and other heavy metals owing to the presence of carboxyl and phosphate groups [94]. Additionally, in *Bacillus* sp. dw-2, the presence of uranium precipitates in the form of small needles has been observed [95]. The denitrifying *Pseudomonas* genus has been distributed with high relative abundance in different bentonite samples [6,85,96], which is well-known for their capacity to interact with uranium through the different mechanisms that contribute to U immobilization. The study of Povedano-Priego et al. discovered the presence of *Desulfovibrio* within the bacterial community of G2P-uranium-treated bentonite under anoxic conditions [96]. The capacity of members of this genus to use glycerol as electron donors had earlier been reported [97]. In this case, the oxidation of glycerol, acetate, or lactate to CO_2_ is coupled to the reduction in U(VI) to U(IV), producing the immobilization of uranium as uraninite [98]. This capacity allows *Desulfovibrio* to survive in uranium-contaminated sediments [99], and waters [100]. 

As in bacteria, the fungal community in bentonite is also affected by the presence of uranium in the environment. This group of microorganisms has not been extensively studied, and there is still a lack of understanding regarding its behavior within the framework of DGRs. In fact, Povedano-Priego et al. found that the fungal diversity in bentonites treated with uranyl nitrate was completely different from U unamended samples, with *Penicillium* and *Fusarium* exhibiting the highest relative abundances in the presence of this radionuclide. High-angle annular dark-field (HAADF) analyses revealed the ability of *F. oxysporum* B1, isolated from uranium-treated microcosms, to generate U-phosphate phases, thereby aiding in the immobilization and detoxification of uranium [101]. Table 2 shows different microorganisms found in bentonite samples with the capacity to interact with this heavy metal.

### 4.2. Selenium

Selenium plays a crucial role as a micronutrient essential for various biological systems, including antioxidant pathways. Nevertheless, whilst beneficial at low concentrations, Se contamination and subsequent bioaccumulation can lead to environmental and human health risks. Se is toxic for bacteria due to its incorporation into sulfur-containing proteins [102]. The presence of Se(IV) impacts the microbial diversity of bentonite, as described by Povedano-Priego et al. They observed that the relative abundance of the archaea *Methanosarcina* decreased significantly in the selenite-treated bentonite microcosms induced by the toxic effect of Se(IV) [85]. However, *Methanosarcina* recovered when the selenite was reduced in the bentonite by members of a bacterial consortium added in the microcosms. The presence of selenium in bentonite may also influence the bacterial communities, leading to an increase in the abundance of bacteria capable of tolerating the metalloid, developing diverse detoxification mechanisms, such as *Pseudomonas*, *Stenotrophomonas*, and *Desulfosporosinus*, amongst others [85]. Several studies have described the reduction in oxidized and soluble forms of Se (selenite and selenate) to insoluble Se(0) in bacteria, such as *Bacillus* [103], *Shewanella* [104], *Stenotrophomonas* [105], and *Pseudomonas* [106]. The reduction process could be carried out through different enzymatic pathways mediated by different reductase activities such as nitrite, sulfite, fumarate, and selenite reductases [23]. In addition, the Se(IV) reduction could be mediated by molecules containing reduced thiol groups (-SH) such as glutathione (GSH). Pinel-Cabello et al. (2021) detected an increase in the presence of glutathione reductase, glutathione-disulfide reductase, and thioredoxin-disulfide reductase in *Stenotrophomonas bentonitica* BII-R7 culture treated with Se(IV), demonstrating that the mechanism for Se(IV) reduction is activated in presence of this toxic element [107]. The produced Se(0) by bacteria is accumulated in the extracellular space and form Se nanoparticles (SeNPs) with different allotropy; amorphous, monoclinic, and trigonal selenium (Figure 3; [108]). The transformation of amorphous to monoclinic and trigonal Se may be mediated by proteins excreted by bacteria, although the specific process is still unknown [109]. This process has also been observed in bentonite microcosms treated with selenite and spiked with a bacterial consortium after 6-month anoxic incubation [85]. It was demonstrated for the first time that Se transformation generating Se nanostructures composed by the trigonal phase, the most stable form, was detected in a ternary system (bentonite, microorganisms, and Se). Table 2 summarizes some of the autochthonous bentonite microorganisms that have been reported with the ability to interact with this metalloid selenium.

**Table 2 microorganisms-12-01025-t002:** Microorganisms found in bentonite samples with the capacity to interact with heavy metals and metalloids.

Microorganism	Taxonomic Affiliation	Interaction Mechanism	Metal	Reference
*Amycolatopsis ruanii*	Actinomycetota (Bacteria)	Biomineralization	Uranium	[90]
*Bacillus sphaericus*	Bacillota (Bacteria)	Biosorption	Uranium	[94]
*Bacillus* sp.	Bacillota (Bacteria)	Biosorption and bioaccumulation	Uranium	[95]
*Desulfovibrio vulgaris*	Pseudomonadota (Bacteria)	Bioreduction	Uranium	[98]
*Fusarium oxysporum*	Ascomycota (Fungi)	Biomineralization	Uranium	[101]
*Bacillus selenitireducens*	Bacillota (Bacteria)	Bioreduction	Selenium	[103]
*Shewanella oneidensis*	Pseudomonadota (Bacteria)	Bioreduction	Selenium	[104]
*Stenotrophomonas bentonitica*	Pseudomonadota (Bacteria)	Bioreduction	Selenium	[105]
*Pseudomonas seleniipraecipitans*	Pseudomonadota (Bacteria)	Bioreduction	Selenium	[106]

## 5. Copper Corrosion under Repository Conditions

Extensive research has been conducted on the corrosion of canister materials under controlled experimental conditions. Therefore, it is necessary to extrapolate the experimental data to take into account the fluctuating evolution of chemical, mechanical, and redox conditions that will occur during the repository post-closure period. Based on the report of Landolt et al. [14] regarding the NAGRA design concept, four phases have been identified to occur on the canister surface from the time of repository closure up to millions of years: (1) in the first tens of years, an initial phase will develop characterized by dry and oxic conditions, with high temperatures; (2) second oxic and unsaturated phase would last the first few hundreds of years; (3) third phase of more advanced hundreds of years, wherein oxygen would have already been consumed and an anaerobic phase with unsaturated conditions would be established; and (4) the cold, anoxic, and long-lasting final phase with fully saturated conditions would be established. The evolution and fluctuations in temperature and oxygen concentration projected over the years after the closure of future nuclear repositories are illustrated in Figure 4 [110].

According to the various phases anticipated after repository closure, the primary factors expected to play a significant role in the corrosion of copper canisters would include: the presence/absence of oxygen, radiation (Section 2), and microbiologically influenced corrosion (MIC) [21]. With regard to oxygen, its presence will be limited to the first years of the repository, as it will be consumed by bacterial activity and by the oxidation of minerals present in the different barriers [111,112]. The most plausible reaction expected to occur is the one that results in copper oxide(I) according to the following equation: 4Cu + O_2_ → 2Cu_2_O [21]. However, when these oxic phases come to an end after the total consumption of O_2_, the environment becomes completely anaerobic. In the context of this reducing environment, copper corrosion processes are primarily expected to result from water reduction, wherein copper would react with water molecules, leading to the formation of copper(I) oxides (Cu_2_O) alongside H_2_ [21,113]. However, one of the most critical sources of corrosion would be due to microbial activity. Sulfate-reducing bacteria are frequently responsible for MIC damage, primarily due to the abundance of sulfate in anaerobic environments. This corrosion process can occur through two different routes: an indirect route known as chemical MIC (CMIC), where corrosion is caused by the products resulting from microbial metabolism; and/or a direct route, known as electrical MIC (EMIC), where the transfer of electrons from the metal directly interfaces with the bacteria and it requires biofilm formation [114,115]. The formation of biofilms on the surface of the canister is highly improbable in a repository concept that includes the direct contact of bentonite backfill with the canister [12]. Additionally, copper lacks the energetic potential for SRB to utilize it as a source of electrons for generating energy through dissimilatory sulfate reduction (DSR). Therefore, between the two pathways, CMIC would predominantly engage SRB since this bacterial group utilizes sulfate, not copper, as final electron acceptor in the respiratory process (dissimilatory sulfate reduction). This allows them to obtain energy by releasing HS^−^ (bisulfide) or H_2_S (sulfide), depending on the pH value [20,116]. The copper would corrode due to these biogenic sulfides resulting in the formation of copper sulfides as described by the following equation: 2Cu + HS^−^ + H^+^ → Cu_2_S (s) + H_2_ (g) [117] (Figure 5). Deep underground water sources might contain elevated levels of dissolved sulfate ions, which could support the sustenance of these SRB. Additionally, nutrients could be sourced from natural organic materials within the clay, fracture fluids, or neighboring minerals [21,118].

To date, there is robust evidence indicating that the requisite conditions for facilitating localized corrosion phenomena, such as pitting, are unlikely to take place in a post-closure nuclear repository using copper as a corrosion barrier [119,120,121,122]. Furthermore, it should be noted that this sulfur flow would occur during the later stages of the DGR, once water saturation has taken place. By this time, it is anticipated that O_2_ consumption will have been completed, thus bringing an end to the oxic corrosion mentioned earlier. Consequently, when sulfide becomes available, the copper would probably encounter a layer composed of copper oxides or hydroxides already formed, rather than pure copper. Smith et al. [123] documented the conversion of Cu_2_O to Cu_2_S under aqueous sulfide conditions (HS^-^), and similar reactions involving other copper species such as Cu^+^ or CuCl_2_ have been reported in other studies. Therefore, the significance of these transformations lies in the fact that under repository conditions, the incoming sulfide will interact with corrosion products such as copper oxides or dissolved copper that have already formed. However, this corrosion product layer prior to sulfide exposure can serve as a barrier, redirecting sulfide to react with the pre-existing corroded material rather than with new canister sections [21].

## 6. Effect of Oxygen, Gasses, and Nutrients on the Microorganisms

With the final closure of the repository, the initial oxic conditions would lead to a complete anoxic environment with time. Initially, oxygen can be consumed by different processes including corrosion of the canister [14], reduced components of the rocks and backfill [124], oxidation of bentonite Fe containing minerals [125], and microbial activity in bentonite [125]. The necessary time to re-establish the anoxic conditions depends on the above-mentioned processes, ranging from months to thousands of years [12]. For example, Giroud et al. [125] considered gas exchange with bentonite pore water and adsorption on mineral surfaces as the most important processes controlling O_2_, with it being fully consumed after only a few months. However, according to Wersin et al. [126], anoxic conditions would be re-established in the buffer after a period ranging between 7 and 290 years, whilst Grandia et al. [127] predicted that aerobic conditions may prevail for more than 5000 years in the absence of O_2_ consumption in the buffer. As far as the microbial involvement in the process is concerned, several authors have confirmed the short term of O_2_ consumption [12]. Apart from the oxygen trapped in the pore spaces, the radiolysis of humid air or water also produces oxidizing species that will increase the general metal corrosion rates, as previously mentioned [128]. The effect of the radiolysis of water on corrosion, however, can be significantly attenuated in the absence of oxygen [129].

Moreover, a significant volume of other gasses would be generated with time in the DGR which may affect the long-term safety of the system by shifting the containment functions of engineered and natural barriers through over-pressurization of the repository. In addition, the hydraulic and mechanical properties of engineered barriers and host rock might be altered [130,131]. Gas would be generated through several abiotic processes such as anaerobic corrosion of metal canister and ferrous components in the engineered barrier system, biotic degradation of organic materials and radiolysis of water [130]. Hydrogen is expected to be the dominant gas generated in a deep geological repository [132,133] and could also potentially act as a carrier for radioactive gaseous species (e.g., H-3, C-14, Rn-222, etc. [134,135]). The formation of H_2_ by anaerobic corrosion of metals may further contribute to gas generation (such as CH_4_, H_2_S or CO_2_), as H_2_ can be used as an electron donor in microbial processes [136]. Soluble organic degradation products (e.g., formic acid, acetic acid or methanol, among others) also have the potential to enhance corrosion and form aqueous complexes with radionuclides, which may affect their mobility and release from the repository [137]. Methane and carbon dioxide are generated as a result of microbiological degradation of organic materials, which could lead to overpressure in the repository and migration of water-borne radionuclides in fractures of crystalline bedrock and may drive the transport of radionuclide contaminated groundwater to the biosphere [137]. In addition, irradiated metals and waste materials may contain radionuclides, which can be volatilized to the biosphere in the form of gas [138]. Although it is assumed that gas-generating materials are not altered prior to the operational and post-closure repository phase, the environmental conditions which control the gas production (e.g., temperature, concentration of reactants, availability of catalysts, salinity, pH, redox potential) may increase or decrease it. Generally, production, transport and consumption of gas and water in a closed deep geological repository are coupled processes. Gas and water transport affect the production/consumption of water and gas, e.g., by controlling the gas pressure and the availability of water for corrosion and degradation reactions, whilst water and gas production/consumption will influence the movement of water and gas [139].

Another important factor to consider is the extreme oligotrophic conditions deep underground in the repository. Life in the continental deep biosphere is broadly constrained by energy and nutrient availability [140]. However, it has been demonstrated that deep biosphere microorganisms are active, and able to influence the environmental conditions and material integrity in deep geological repositories [93,141]. These extreme and deep ecosystems foster diverse, yet cooperative, communities adapted to this setting [142]. Metabolic cooperation, via syntropy between sub-surface microbial groups, is critical for the survival of the whole community under the oligotrophic conditions that dominate in the sub-surface, where microbial ecosystems are typically supported by H_2_. Methanogens such as *Methanosarcina*, *Methanoculleus*, and *Methanocella*, and sulfate reducers like *Desulfosporosinus*, *Desulfovibrio*, and *Desulfotomaculum*, and the respective energy processes, are thought to be the dominant players [143]. Furthermore, the main source of microorganisms in the bentonite appears to be the bentonite itself rather than the host rock or its pore water. The number of microorganisms is predicted to increase until full saturation of bentonite is reached and would remain at stable numbers for decades [141]. Interestingly, Burzan et al. [73] observed the growth of aerobic heterotrophs instead of anaerobes in the bentonite despite the nominally anoxic conditions. The microbial activity of anaerobic microorganisms (most commonly sulfate-reducing bacteria) can lead to corrosion of the metallic waste canister materials [73], and the production of gasses such as methane, hydrogen sulfide, and nitrogen [144], affecting the long-term safety of the repository. Therefore, taxonomically and metabolically diverse microorganisms developed syntrophic partnerships to actively overcome all the difficulties imposed by the environmental conditions in the deep sub-surface, and thus, affect the safety of the DGRs for nuclear waste.

## 7. Effect of Temperature

### 7.1. Impact of Temperature Evolution on Nuclear Repository Barriers

The waste disposal temperature will be shaped by the layout, design, and decay of radionuclides within the spent nuclear fuel bundles in the DGR [112]. Then, the subsequent evolution of the temperature over the storage time would be predominantly determined by the thermal behavior of the canisters due to the heat generated by the decay of the radionuclides. Different models based on the DGR designs of each country predict a gradual increase in the temperature in the canister’s surface, reaching a maximum peak of ≈100 °C within the first decades (Table 1). Additionally, this generated heat would dissipate into the nearby engineering barriers and the host rock. The temperature evolution is a key factor that may alter the properties and stability of engineered barriers such as the bentonite sealing material or the metal canisters. Since the DGR environment would not be sterile, the temperature evolution could affect the microbial communities within the repository.

Elevated temperatures can compromise the mineralogical, mechanical, and index characteristics of bentonite. Kale and Ravi [145] reported that, with an increase in the temperature, the liquid limit (the water content where the soil starts to behave as a liquid) of the bentonite decreases. This is also affected by the clay content, since the higher the clay content, the higher percentage reduction in the liquid limit. This decrease in the liquid limit, with a temperature rise, would affect the plasticity (ability to undergo deformation without cracking) index in the same way. Moreover, the specific gravity (density of a substance in comparison to the density of water) of smectite soil was found to be reduced by the increase in the temperature. Based on the results of Tan et al. [146], if the temperature reaches >100 °C, the specific gravity will decrease gradually up to 400 °C followed by a slow decrease up to 800 °C. Additionally, the maximum dry density increases with an increase in the temperature [145]. Exposure to high temperatures results in several physical and chemical transformations in clay, such as mineral composition/decomposition, reduction in moisture layers, and alterations in mass and density, contributing to the degradation of its structural integrity [147]. The water content is influenced by the temperature since water layers begin to reduce their size when the clay is in contact with heat [148]. The moisture layers can be removed when the temperature is between 100 °C and 110 °C (absorbed moisture) leading to the collapse of the interlayer pore space. The optimum moisture content is less affected at <100 °C, whilst up to 400 °C, a drastic decrease can be observed. Subsequently, this loss of water affects the macro and microporosity of the clay minerals and their plasticity. The physicochemical properties of bentonite can be also affected by an increase in temperature. Estabragh et al. [149] showed a gradual decrease in Na^+^ concentration, linked with an increase in temperature, affecting the cation exchange capacity (CEC) of the bentonite, whilst a pH decrease was evidenced when the temperature was >150 °C. When clay is heated, the bonds at the edges of the interlayer units, especially in montmorillonite, will be broken due to the presence of weak Van der Waals forces. At a mineralogical level, one of the main concerns related to elevated temperatures is the irreversible smectite-to-illite transformation that affects the swelling capacity of the bentonite [150]. This process, known as illitization, involves the chemical alteration of expandable smectite into non-expandable illite. This change is typically prompted by elevated temperature exposure (100–200 °C) over a long period of time [151]. The process of illitization may be initiated by potassium ions, potentially originating from the clay K-feldspars, which integrate into the smectite layers and form covalent bonds with oxygen [150].

Moreover, in this context, published studies show that shifts in temperature have a negative effect on metal canister corrosion affecting the corrosion kinetics and the solubility of mineral phases in contact with the canisters [12]. The corrosion rates of passive metals are remarkably low (around 0.01 µm/year). However, the integrity of passive self-forming surface films faces threats from localized corrosion, including pitting and crevice corrosion, with temperature playing an important role in initiating corrosion [152]. A suitable material for canisters must preserve its structural integrity, corrosion resistance, and radiation shielding capabilities at temperatures > 100 °C, potentially up to an upper threshold of 300 °C (Table 1). Hence, the concern that sensitized microstructures susceptible to localized corrosion may be produced, compromising the mechanical integrity of the canisters, does not appear to be an important issue under the temperatures expected in a DGR. Due to the heat on the surface of the canisters, the initial humidity in the bentonite would be distributed to cooler areas. This would result in dry and uniform oxidation on the surface of the canisters. Studies conducted by Stoulil et al. [153] have shown that high temperatures compact the corrosion layers on the surface of the canisters, which slightly reduces the corrosion rate. Schlegel et al. [154] showed that under fluctuating temperature conditions (85 °C with a drop to 25 °C, and then 85 °C again), steel corrosion presented areas of extensive corrosion with depressions filled with corrosive products. This fluctuating temperature produced a high rate of corrosion compared to when the temperature was static at 85 °C. Moreover, in conditions where the temperature varies, the formation of carbonates, chlorides, sulfides, Fe-silicates, and magnetite as corrosive products are common. Schlegel et al. [154] determined that significant effects of low-temperature fluctuations include accelerated metal oxidation compared to the formation of corrosive products [155], growth of microbial colonies leading to biofilms or Fe-sulfide generation, and a decrease in air tightness. However, with a rise in temperature, corrosive product deposition intensifies, which in turn decelerates the corrosion process. In the case of copper, the main corrosive agent under anaerobic conditions is hydrogen sulfide produced by SRB [77]. Martinez-Moreno et al. [69] showed that these bacteria do not exhibit viability at high temperatures (60 °C). Therefore, no precipitates related to copper anaerobic corrosion (Cu_x_S) were observed, with copper oxide being the main corrosive product detected.

### 7.2. Impact of Temperature Evolution on Microorganisms

One of the aspects to consider when ensuring the stability of a DGR is the effect that microorganisms will have on the different containment barriers under repository-relevant conditions [156]. The elevated temperature could affect the microbial communities within the DGR, especially those naturally occurring in the bentonite. Martinez-Moreno et al. [69] proved that high temperature (60 °C) reduces the microbial diversity and the number of SRB in compacted bentonite after a one-year incubation. Moreover, at this temperature, whilst some aerobic bacteria (e.g., *Aeribacillus*) showed growth capacity at 60 °C, certain anaerobic bacteria (IRB and SRB) did not. The activity absence of IRB and SRB at 60 °C, or higher temperatures, is of particular interest related to the MIC of metal canisters previously mentioned, as this fact would retard the MIC on the canister surface. Additionally, Bartak et al. [157] found a decline in microbial activity as temperature rose, and that thermal treatment significantly impacted the microbial community composition within the bentonite, identifying various thermophilic species (e.g., *Caldinitratiruptor* and *Brockia*) or thermotolerant spore-forming genera (e.g., *Thermincola* and *Bacillus*). Moreover, they determined a threshold temperature of 90 °C to inhibit microbial activity and growth across all the tested bentonite suspensions. Furthermore, previous research has demonstrated that bacteria can withstand extreme environments either through the formation of spores (e.g., *Desulfosporosinus* and *Desulfotomaculum*), or by entering into a dormant state, such as desiccated cells [158,159]. Since high temperatures turn the bentonite/canister interface into a dry environment, bacteria could face water loss enhanced by the strong affinity of the bentonite, which draws water away from the cells causing cell inactivation and inducing spore formation. Moreover, the formation of biofilms and sporulation are well known tactics used by bacteria to withstand desiccation [160]. Once the temperature starts to be cooler during the anoxic period (Figure 4), bacteria can recover their cell viability under favorable conditions [69,159].

## 8. Limitations, Challenges, and Opportunities

Ensuring the long-term stability of future nuclear waste repositories requires very comprehensive studies based on multidisciplinary approaches, combining engineering, chemistry, physics, mathematics, geology, and microbiology. The effect of mineralogical and biogeochemical parameters (e.g., clay mineralogy and compaction density, pore water chemistry, etc.) on the stability of DGR barriers have been well studied. However, very few studies were conducted to investigate the combined impact of different DGR physicochemical parameters (e.g., radiation, temperature) on the presence and activity of microbes. Incorporating these microbiological data in the geochemical models is of great importance to achieve an accurate prediction of the long-term stability and safety of the different DGR barriers. Nevertheless, this topic is in its infancy and close collaborations between experimentalists (microbiologists, chemists, mineralogists, physicists, etc.) and modelers should be conducted to fill the gaps. Introducing bacterial activity into long-term mathematical models presents a challenge due to the inherent complexity of biological systems and the interactions between their components (e.g., complexity and diversity of bacterial behaviors, dynamic and nonlinear interactions, variability and heterogeneity within the bacterial community, etc.). Therefore, the principal challenge facing the scientists working in this field is the complexity of these studies as they require taking into consideration the physiochemical and microbiological relevant conditions for future DGR (temperature, radiation, water activity, etc.) and a long-term duration of the research projects. Therefore, in order to simulate a more realistic scenario, this kind of studies should be conducted at the underground laboratories such as Mont Terri Rock (Switzerland), Hades (Belgium), Äspö Hard Rock (Sweden), and Meuse/Haute Marne (France). The access to such laboratories is, however, very limited. Advance in this field of study is crucial and should be based on the communication between experts in the relevant disciplines. This will enable the collection of comprehensive, realistic data and the development of accurate strategies to guarantee the long-term safety of these systems.

## 9. Conclusions

The aim of this review is to outline how various conditions evolving over the lifespan of a nuclear waste repository can impact the properties of engineered barriers (metal canister and backfill/seal barrier). Key parameters influencing the safety of these repositories include radiation, temperature fluctuations, bentonite compaction density, and abiotic/biotic factors (e.g., oxic/anoxic atmosphere, nutrient-enriched groundwater seepage, gas production), amongst others. We have also discussed the impact of microbial processes on the safety performance of DGR barriers. Despite the harsh conditions involved within a DGR that may impact the survival and viability of microorganisms, some autochthonous microbes from bentonite and those that will be accidentally introduced due to human activity may be inhabiting there. Hence, understanding the interaction between microorganisms and the barriers is crucial since they are able to completely modify the surrounding environment. For this reason, research on the microbiology of repositories has been increasing in recent years. Moreover, within this review, we have highlighted the behavior of autochthonous bentonite microorganisms in response to radionuclide leaks from the nuclear waste, such as U or Se, underscoring the capacity of certain microbes to retard their mobility and migration through bioimmobilization processes. In this context, understanding the conditions that could impact the different barriers is crucial for ensuring the long-term safety of future DGRs, since these systems will be highly complex.

## Figures and Tables

**Figure 1 microorganisms-12-01025-f001:**
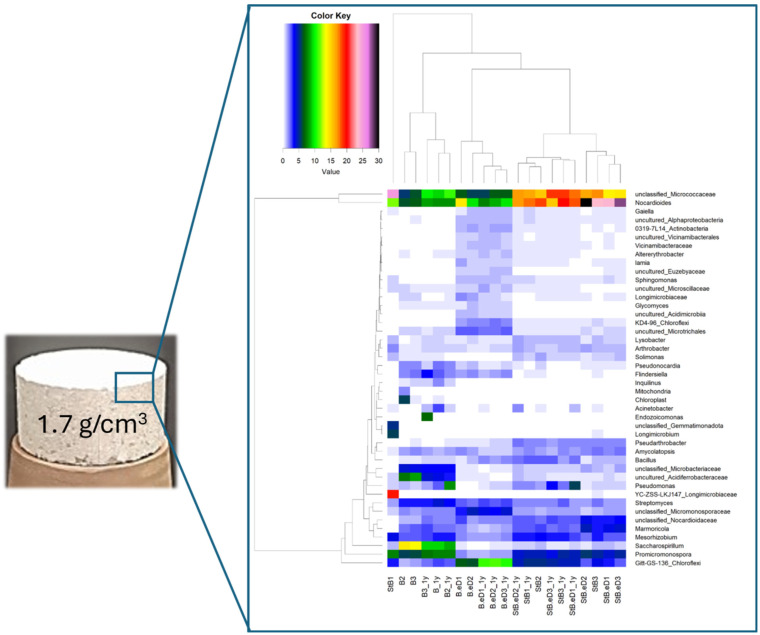
A Spanish bentonite block compacted at a dry density of 1.7 g cm^−3^. Heatmap of the relative abundance of the samples at genus level in triplicate (duplicates in StB.eD, B, and B.eD). Cut-off: 0.5% of r.a. Different colors show the relative abundance of each genus (the warmer the color, the greater relative abundance). Data from Martinez-Moreno et al. (2023) [68].

**Figure 2 microorganisms-12-01025-f002:**
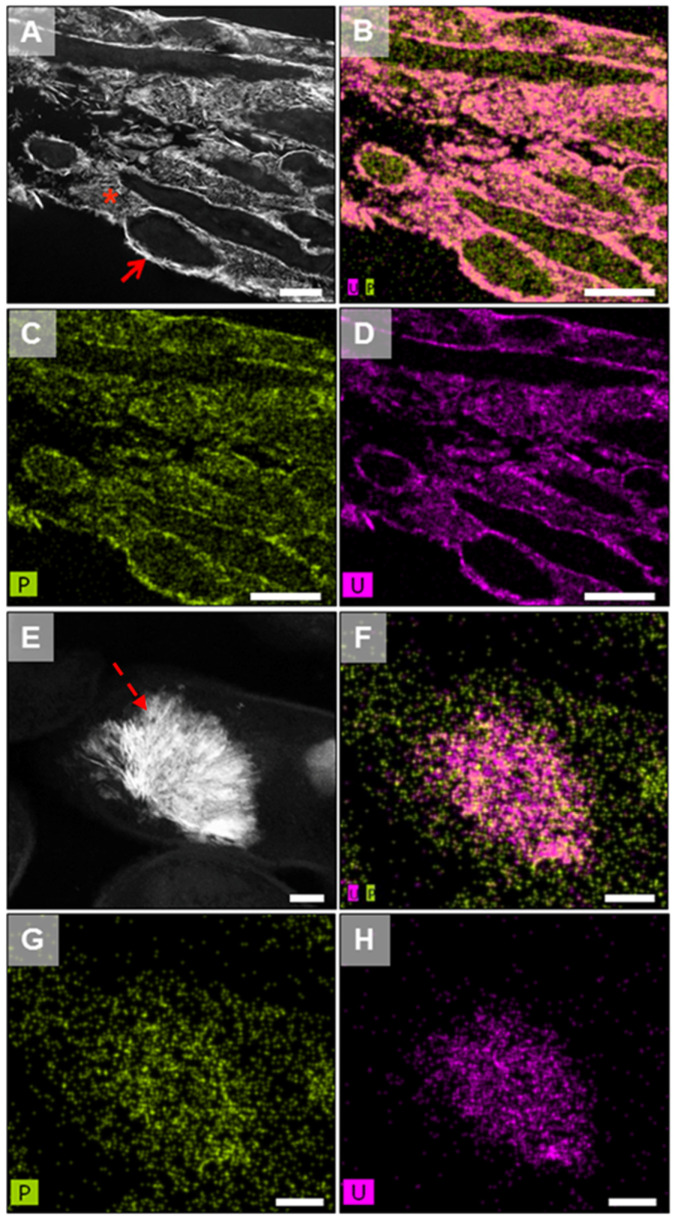
High-angle annular dark-field scanning transmission electron microscopy (STEM-HAADF) images of thin sections of *Amycolatopsis ruanii* cells treated with uranium and glycerol-2-phosphate (G2P) showing U-P deposits at cell wall level (arrow), extra- (asterisk) and intracellular (dashed arrow) uranium phosphates (**A**,**E**), and their corresponding EDX maps with the distribution of P (**C**,**G**), U (**D**,**H**), and P + U (**B**,**F**). Bar scale: 500 nm (**A**); 800 nm (**B**–**D**); 100 nm (**E**–**H**). Figure from Povedano-Priego et al. [90].

**Figure 3 microorganisms-12-01025-f003:**
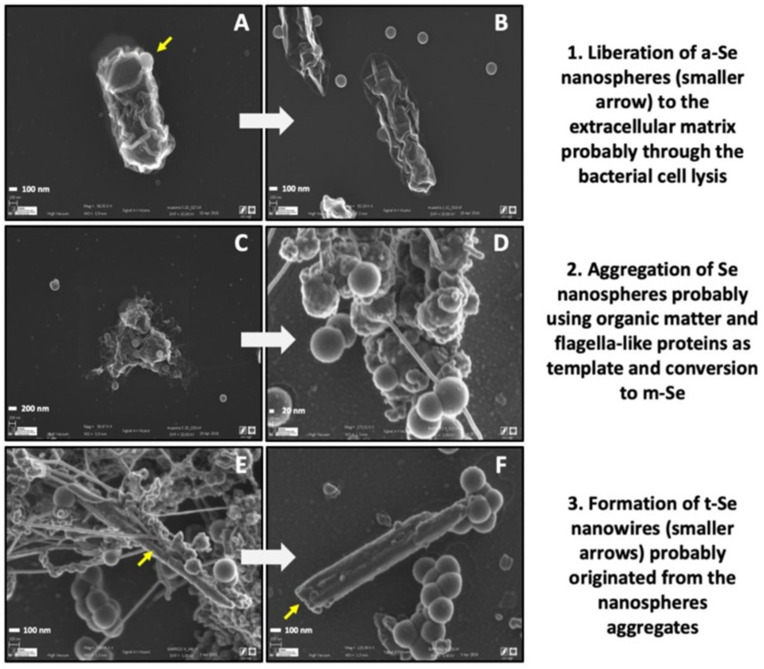
VP-FESEM images illustrating the Se transformation from a-Se nanospheres to t-Se nanowires, with an intermediate step of m-Se aggregates by using proteins as a template. The images correspond to samples prepared by growing *Stenotrophomonas bentonitica* anaerobically. Scale bars: 100 nm (**A**,**B**,**E**,**F**), 200 nm (**C**), and 20 nm (**D**). Yellow arrow in (**A**): *a*-Se nanospheres; yellow arrow in (**E**,**F**): *t*-Se nanowires. Figure from Ruiz-Fresneda et al. [108].

**Figure 4 microorganisms-12-01025-f004:**
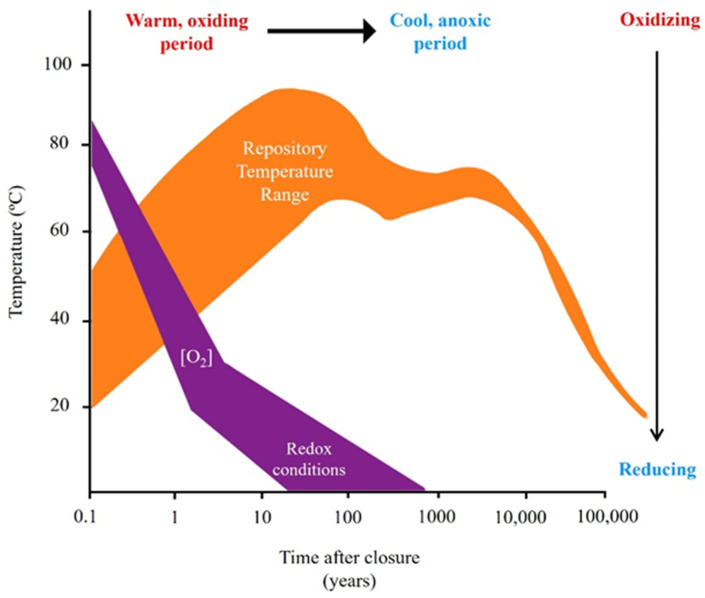
Diagrammatic representation of the evolution of the near-field environment for a repository. Modified from [110].

**Figure 5 microorganisms-12-01025-f005:**
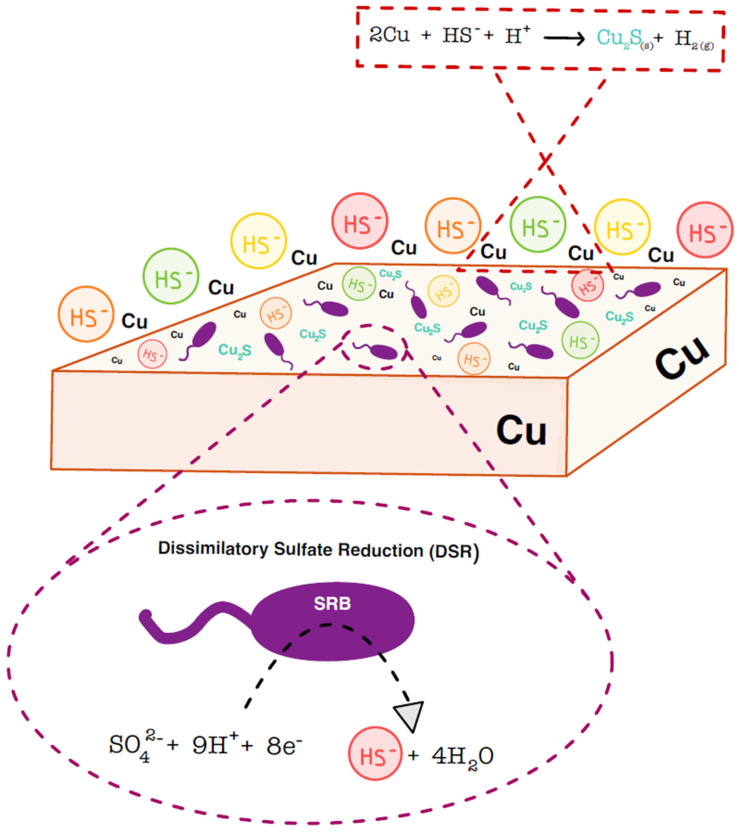
Schematic representation of chemical microbially induced corrosion (CMIC) of copper by sulfate-reducing bacteria (SRB).

**Table 1 microorganisms-12-01025-t001:** Information on DGR models to be followed by each company representing the main countries involved in the management of high-level nuclear waste.

Country	Company	Canister	Buffer	Buffer Density (g/cm^3^)	Host Rock	Absorbed Dose at the Surface (Gy/h)	Temperature at the Surface (°C)	References
Spain	ENRESA	Carbon steel	Bentonite	1.65	Clay/Granite	Not determined	<100	[7,8]
Finland	POSIVA	Copper + cast iron	Bentonite	1.55	Crystalline	0.33	~90	[9,10,11]
Sweden	SKB	Copper + cast iron	Bentonite	1.6	Crystalline	0.2	~90	[10,12]
Switzerland	NAGRA	Carbon steel	Bentonite	>1.45	Opalinus clay	<0.035	<150	[10,13,14]
France	ANDRA	Carbon steel	none	-	Granite	<10	~90	[10,12]
Czech Republic	SÚRAO	Carbon steel	Bentonite	1.4	Crystalline	0.3	<95	[12,15,16]
Belgium	ONDRAF-NIRAS	Carbon steel	Cement/c	-	Boom clay	25	~95	[10,12]
Canada	NWMO	Carbon steel coated with copper	Bentonite	1.6	Crystalline/sedimentary	2	<100	[17,18,19]

## Data Availability

No new data were created or analyzed in this study. Data sharing is not applicable to this article.

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
