# Peer review of "Insights into the Impact of Physicochemical and Microbiological Parameters on the Safety Performance of Deep Geological Repositories"

_microorganisms, 2024, doi:10.3390/microorganisms12051025_

Round 1

Reviewer 1 Report (Previous Reviewer 1)

Comments and Suggestions for Authors

Dear Authors,

Thank you for updating the manuscript. However, it contains several mistakes. The reviewer suggested not using bulk references, checking reference citations properly, and minimizing text. However, the authors did not carefully consider these. 

For example, please help me understand the references in Table 1. I went through these references, and I did not find the exact values in this table. Please use one row for the values and cite appropriate references only.  

"Microorganisms have the potential to affect their surrounding environment and, consequently, the safety of DGRs through various processes. These include the generation of gases, corrosion of metal canisters, alteration of redox conditions, transformation of mineral clays, and interaction with radionuclides [22, 23, 24, 25, 26]." Please help me understand which paper has mainly highlighted this statement. 

"This review will specifically focus on its impact on bentonite clay's chemical and mineralogical composition, as well as its cation exchange capacity [40, 41, 42]". Please check it.  

"Therefore, one of the essential requirements for its utilization as a buffer and  sealing material would be the rheological and chemical stability when exposed to ionizing  radiation and even in the presence of radionuclides in the worst-case scenario of a waste  leak [43, 44, 45, 46, 47, 48]." Please check.

"The literature on bacterial species highly resistant to radiation is quite limited; however, two strains are referenced and have been well studied for their high resistance to radiation, namely Deinococcus radiodurans, with a tolerance of up to 17 kGy, and Kineococcus radiotolerans [61, 62, 63, 64, 65]". Please check the meaning and references.

"Nevertheless, in the future repositories, microorganisms will not only face radiation as a stress factor [55]. Therefore, more research is needed to understand the evolution of microbial communities under the combination of various repository conditions". 

I request that the authors minimize the text and re-arrange the paper for publication (Abstract, 1. Introduction, 2. Effect of radiation (2.1. Copper, 2.2. Bentonite (2.2.1. Effect of bentonite's density in DHRs), 2.3. Microbial), 2.4. Effects of Radionuclides, (2.4.1. U, 2.4.2. Se, 2.4.3. Cu), 2.5. Effect of gasses and nutrients on microorganisms, 2.6. Effect of Temperature...... 2.7. Discussions, 3. Limitations, Challenges, and Opportunities. 4. Conclusions. 

The paper contains many mistakes. Please carefully revise it. After a major revision, I may recommend your paper. 

Author Response

Reviewer 2 Report (Previous Reviewer 2)

Comments and Suggestions for Authors

Insights into the impact of physicochemical and microbiological parameters on the safety  performance of Deep Geological Repositories

The authors have addressed all of the comments. I have no additional comments to make. The paper may be considered for publication.

Round 2

Reviewer 1 Report (Previous Reviewer 1)

Comments and Suggestions for Authors

Dear authors,

Please consider revising the bulk references and references in Table 1 and correcting them as suggested in the Reviewer's previous comments. This correction can enhance the reliability of the paper. Thank you.

Author Response

The authors have carefully reviewed the references as suggested by the reviewer #1. After the exhaustive revision, we have removed some of the bulk references and kept the most necessary ones. The authors agree that no more references could be deleted. Due to the removal of references, the reference numbers have been rearranged. All changes in the minor revision are highlighted in pink. We hope that these changes are sufficient to address the reviewer's suggestions. 

This manuscript is a resubmission of an earlier submission. The following is a list of the peer review reports and author responses from that submission.

Round 1

Reviewer 1 Report

Comments and Suggestions for Authors

Title: "Unveiling the Unseen: Exploring the Influence of Physicochemical and Microbiological Parameters on the Safety Performance of Deep Geological Repositories"

Comments:

1. Please include each author's affiliation details, email ID, and corresponding authors.

2. Line 21: It is suggested that isotopes of radionuclides be included. Is Se radioactive?

3. Line 81: What does SF stand for?

4. "Unveiling the Unseen": The reviewer discerns no novel message in this review article. It summarizes accumulated findings from the literature without highlighting any research gap. Much of the section revisits topics already discussed in the introduction.

5. The reviewer suggests incorporating more tables rather than extensive text in this article.

6. Additionally, the reviewer recommends avoiding excessive referencing within sentences. It would be preferable to emphasize the key findings of each reference.

7. A recently published article (https://doi.org/10.3389/fmicb.2023.1134078) explored the impact of microbial processes on the safety of deep geological repositories for radioactive waste. It is suggested that the advancements made in this review be compared.

9. Please include Limitations, Challenges, and Opportunities.

10. Furthermore, whether this review article is suitable for publication in this journal is doubtful.

Therefore, I recommend against its publication here.

Reviewer 2 Report

Comments and Suggestions for Authors

Discovering the unseen: insights into the impact of physico chemical and microbiological parameters on the safety performance of Deep Geological Repositories

The article is interesting, aiming to provide an overview of how different conditions that occur throughout the lifespan of a nuclear waste repository can affect the properties of engineered barriers. Several factors play a crucial role in determining the safety of these repositories. These factors include radiation levels, temperature fluctuations, the density of bentonite compaction, and various abiotic and biotic factors.

Few suggestions to the authors.

·         Highly compacted bentonite blocks are thought to prevent microbial growth- Do bentonite blocks have long-term antimicrobial properties?

·         A wide variety of physical 61 (temperature, radiation, groundwater filtration, bentonite compaction, etc.), chemical 62 (gaseous compounds, corrosion, presence/absence of oxygen, etc.), and biological factors- Please provide one example for each factor category.

·         Any evidence of metals that have undergone oxidation/reduction processes as a result of microbial exchange of electrons and protons.

·         Which species of methanogens were the most prevalent? mention some specific examples.